

# Isotopic exchangeability reveals that soil phosphate is mobilised by carboxylate anions whereas acidification had the reverse effect

Siobhan Staunton[1], Chiara Pistocchi[1]

[1]Eco&Sols, INRAE-IRD-Cirad-Institut Agro Montpellier-University of Montpellier, 34060 Montpellier, France

*Correspondence to*: Siobhan Staunton (siobhan.staunton@inrae.fr)

**Abstract.** Mineral P is an increasingly scarce resource and therefore the mobilisation of legacy soil P must be optimised to maintain soil fertility. We have used isotopic exchangeability to probe the lability of native soil P in four contrasting soils following acidification and the addition of carboxylate anions (citrate and oxalate) in soil suspension. Acidification tended to cause immobilisation of soil P, but this was attributed to a salt effect. Addition of both citrate and oxalate led to marked increases in mobilisation of soil P. This would result from both competition between carboxylate and phosphate ions at adsorption sites and chelation of charge compensating cations. The carboxylate effects were similar at each level of acidification, indicating that effects were largely additive. This is not true for the most calcareous soil where calcium oxalate may have been precipitated at the highest oxalate addition. Promoting carboxylate anions in soil by soil amendment or the use of crops that exude large amounts of such organic anions is a promising approach to improve soil P availability.



## 1 Introduction

Phosphorus (P) is an essential nutrient. The bioavailability of soil P is severely restricted because only orthophosphate in solution, accounting for less than 1% of soil P, is directly available for uptake. Inorganic P is largely adsorbed on soil clay minerals and metal (oxy)hydroxides or precipitated. Highly specific adsorption takes place on anion exchange sites of mineral surfaces, in particular clay edge sites, and metal oxides, both as discrete particles and surface coatings, and is characterised by marked hysteresis (Arai and Sparks, 2007; Crisler et al., 2020; García et al., 2021; Gérard, 2016; Geng et al., 2022). Mineral P is a limited resource. The use of P-fertilizer must be rationed and more sustainable approaches to P nutrition found. The research focus has thus shifted from fertilization to the optimisation of legacy soil P, i.e. P stored in soils with a long fertilization history. The challenge for the 21$^{st}$ century is to find strategies to mobilise soil legacy P with plant breeding and the judicious choice of soil amendments. Agronomic strategies include reasoned choice of crop varieties, promotion of microbial activity, addition of immobilised enzymes, pH adjustment, including by compost amendment, and soil enrichment with carboxylate anions, including citrus residues (Alam et al., 2023; Jalali and Jalali, 2022; Paredes et al., 2022; Sindhu et al., 2022). Inspiration for effective soil amendments depends on an understanding of the forms and exchangeability of soil P combined with observation of the strategies of plants and soil biota to cope with P deficiency (Soumya et al., 2022)

Among the most widely considered modifications thought to promote soil P availability are the modification of pH and the enrichment in carboxylate anions. The mechanisms of solubilization of P by acidification and organic anions are quite well understood and have been reviewed (Geng et al., 2022; Barrow, 2017; Duputel et al., 2013; Khademi et al., 2009). The forms of orthophosphate in soil and the charge on P adsorption sites are pH-sensitive, leading to pH-sensitivity of adsorption-desorption reactions. It is now accepted that acidification and carboxylate release by plants and soil biota are separate processes. Carboxylates may exchange with P on soil mineral surfaces and chelate metal cations thus solubilising P-carrier phases. The extent of the role of carboxylates on P nutrition *in situ* remains poorly quantified and lies in the difficulty in quantifying their release by plants. One of the reasons is that the carboxylates are strongly adsorbed by soil, and so quantification depends on their desorption. They are also short-lived due to biological breakdown. To avoid this, studies are often carried out in solution or in an inert matrix such a quartz, but it is recognised that plant physiology may be very different in soil (Wang and Lambers, 2019).

The relation between P solubility and pH is complex; acidification does not always promote solubility. For example, the mechanism of action of phosphate solubilising bacteria may not be acid release, as it is often



presumed (Biswas et al., 2022; Barrow and Lambers, 2022; Menezes-Blackburn et al., 2016). The maximum of P
availability is often assumed to occur at neutral pH (Penn & Camberato, 2019). This is attributed to the
combination of pH dependences of P fixation by iron, aluminium and calcium. However, both the explanation and
the assumption are contested by Barrow (2017), who reported the minimum in the U-shaped relation between P
desorption and pH at about pH 5. In calcareous soils, carbonates may be dissolved in preference to sparingly
soluble phosphates upon acidification (Geng et al., 2022). The relative effects of pH and carboxylate anion are
therefore not easily predicted, not least because some studies investigate carboxylate by adding the acid form.
Another important question is how to assess P availability reliably. Although the objective of all studies is to
understand and predict biological P availability, uptake is species-dependent and so chemical extractions are often
preferred. Common approaches assess the size of operationally defined "pools" of soil P. Water-extractable P is
usually too low to be reliably measured, whereas harsh extractants (e.g., 1M HCl) are unlikely to detect small
changes in P availability. Furthermore, the size of soil P pools is not related to their capacity to release P and the
rate of this release (Helfenstein et al., 2020). Other common approaches to understanding and predicting soil P
dynamics include the study of adsorption isotherms. However, adsorption measures the affinity of soil to adsorb
additional, often environmentally unrealistic, amounts of P. Adsorption isotherms give no information on the
lability of either native or recently added P. Methods that probe the dynamics of immobilised soil P should be
better suited to measure P availability. This is because biological uptake creates a depletion gradient and so
availability is determined by the ease of replenishment of solution, namely the lability of immobilised P. One
potentially suitable approach is resin-extractable P, which is reputed to be a good proxy for P that could be taken
up by growing plants. Isotopic exchangeability provides another approach, with the advantage that it probes soil
P availability without displacing the sorption-desorption equilibria (McBeath et al, 2009 and references therein).
Combinations of chemical extraction, spectroscopic techniques and isotopic exchange suggest that they provide
coherent information on P status in soil (Braun et al., 2020; Helfenstein et al., 2018; Randriamanantsoa et al.,

69   2013).

The aim of this study was to elucidate the separate effects of pH and carboxylate anions on the solubility of
phosphate. Four contrasting soils, with different pH, P availability and carbonate content, were studied. We
focussed on the effect of two carboxylates reported to be released by plants and associated microorganisms
(Bhattacharyya et al., 2011), citrate and oxalate. These anions have been found to have the largest effect on P
solubility (Staunton and Leprince, 1996). We chose to use short-term (2-h) isotopic exchange of native soil P to
assess its lability.



## 2 Materials and methods

Four contrasting soils were selected for this study and some properties are given in Table 1. Soils were sampled in the Mediterranean region; one was acid, another neutral and two were calcareous. They also differed in their P-status and pH-buffer capacity.

Soils were air-dried, sieved to 2 mm and stored before use. All measurements were carried out in suspension (2 g soil/ 4 ml). Preliminary experiments were carried out to measure soils pH and pH buffer capacity by addition of HCl. Most soils showed a dramatic drop in pH when acidification was greater than 1 unit, and so we decided to limit pH adjustment to this value. Preliminary tests showed that both carboxylate anions strongly decreased the adsorption of added phosphate. Water extractable P (using the molybdate blue method) was found to be below the detection limit for three of the soils and about 3.6 µM for Montana. pH was measured on the aqueous phase of suspension after settling of the suspension. The measurement of isotopically exchangeable P was carried out by adding carrier-free $^{32}P$ to each suspension to obtain a final content of 50 MBq/ml of. Soils were suspended in solutions containing strong acid (HCl) to acidify the pH by about 0.5 or 1 pH unit, and /or the sodium salt of oxalate or citrate ($10^{-4}$, $10^{-3}$ or $10^{-2}$ N), or NaCl ($10^{-2}$ N), as required. NaCl was added to test for a possible salt effect independent of the chelating action of carboxylate, i.e., change in ionic strength and therefore surface charge density and formation of cationic bridges. Since both citrate and oxalate are strongly adsorbed, no attempt was made to quantify the proportion of anion adsorbed, nor to follow the kinetics of microbial breakdown of the anion. After a contact period of 2 hours between soil and solution, phases were separated by double centrifugation (10 minutes at 19 000 $g$, followed by 30 minutes at 19 000 $g$ of 2 ml aliquots of the first supernatant solution). The proportion of $^{32}P$ adsorbed was calculated from the difference of $^{32}P$ content in the initial solution and after 2-hour contact. All experiments were carried out in triplicate. After phase separation, $^{32}P$ activity was quantified by liquid scintillation, after addition of 4.8 ml scintillation liquid (Aquasafe) to 0.5 ml aliquots of each sample (in triplicate). The distribution coefficient, defined as the ratio of adsorbed ($x_{32P}$) to solution phase concentration of $^{32}P$, $[^{32}P]$, was calculated as follows:

$$\frac{x_{32P}}{[^{32}P]} = \frac{V_L}{M_S} \left( \frac{[^{32}P]0}{[^{32}P]} - 1 \right)$$

where $V_L$ is the volume of solution in suspension and Ms the mass of soil, and the subscript 0 refers to time 0. This parameter is mathematically equivalent to the tangent of an adsorption/desorption isotherm and is inversely proportional to the lability of legacy P and to the effective diffusion coefficient of phosphate (Lambers and Plaxton, 2015).



**3 Results and discussion**
Soils differed in their pH buffer capacities as shown in Table 1. The presence of oxalate or citrate led to small
alkalinisation (about 0.1 unit, not shown). Preliminary experiments with adsorption isotherms showed that addition
of 10 mM citrate or oxalate caused marked decreases in P adsorption, as reported in a similar study (Staunton and
Leprince, 1996), but we decided not to pursue the investigation of P adsorption isotherms, because of the limited
relevance of large amounts of added P on the dynamics of legacy P.
Figure 1 shows the effect of anion addition on the distribution of $^{32}P$ on each of the soils. The addition of $10^{-2}$ N
sodium chloride (20 mmol kg$^{-1}$ soil) is also shown (closed symbols) to distinguish between a true effect of
carboxylate, and a simple salt effect. It also indicates the effect of the addition of chloride from HCl. The salt effect
observed for each of the soils, except Cazevieille, led to decreased mobilisation of legacy P, in contrast to the
effect of carboxylate. For each of the soils, addition of carboxylate anions led to marked decreases in the
distribution coefficient ($x_{32P}/[^{32}P]$), namely mobilisation of legacy P. The threshold for observation of a significant
effect is about 1 mN for oxalate, and somewhat greater for citrate. It should be remembered that solutions had the
same normality, so concentrations of oxalate were larger than those of citrate. Two mechanisms would account
for this effect. Firstly, ligand exchange between carboxylate anions and adsorbed phosphate should liberate P.
Secondly, the chelating effect of the oxyanions solubilise Fe and Al oxides that are important carrier phases for P.
Similar additions of carboxylate anions have been reported to increase P lability (Staunton and Leprince, 1996;
Menezes-Blackburn et al., 2016; Barrow et al., 2018), although, unlike the present study, in some studies weak
acid was added, therefore effects of anion and acidification cannot be separated.
Acidification led to marked decreases in P lability. This apparent acidification effect (calculated at ΔpH of 0.5)
appears to be inversely proportional to initial pH (greatest in the acid soil) and to the average buffer capacity (Fig.
2). The closed symbols on Fig. 1 show the salt effect, with no change in pH, namely the addition of $10^{-2}$ N NaCl.
These observations suggest that the apparent acidification effect is in fact a simple salt effect, for most of the soils.
The salt effect (calculated at ΔpH of 0.5) also appears to be positively related to the pH effect, with the neutral
Cazevieille soil being a marked exception to this trend (Fig. 2). The observation of a salt effect suggests the
implication of sorption on variable charge surfaces, e.g. kaolinite and metal oxides, in P immobilisation (Barrow,
2015). Only in the case of the neutral Cazevieille soil the decrease in P lability could not be explained by a salt
effect. A true pH effect is more complex. The formation of phosphate bridging complexes at clay edge sites and
with the hydroxyl groups of Al-OH and Fe-OH consume protons, and so equilibrium will be displaced by pH
changes. In the present study, the pH effect was not continuous, with an initial increase of $x_{32P}/[^{32}P]$, followed by



a decrease to close to the initial value, for the two calcareous soils, Château Fourques and Restinclière. This could
be interpreted as a U-shaped pH-dependence of P lability, or the result of two different processes with opposite
pH effects. Acidification of calcareous soils dissolves carbonates releasing calcium, explaining the large buffer
index for these soils, and to a lesser extent phosphate from precipitated calcium phosphate. Acidification by 0.5
pH units might not be sufficient to solubilise Ca-phosphates, and so the apparent acid effect is dominated by salt
effect. Further acidification could cause the release of P from Ca-phosphates or co-precipitated with carbonates,
and displace the pH-dependent equilibria of adsorbed P. The acid soil, Montana Colera, was the only soil where
the increasing acidification led to a further (small) increase in $x_{32P}/[^{32}P]$, confirming a trend in addition to a salt
effect. For this acid soil acidification could not have released Ca from carbonates.
It must be recalled that the pH changes will have little effect on the acid-base equilibria of the carboxylate anions.
Oxalate (pKa = 1.23 and 4.28) would be in divalent form at all pH. The equilibrium between the divalent and
trivalent forms of citrate (pKa = 3.13, 4.76, 6.39) would change over the pH range, with roughly equal amounts of
both around pH 6.4; more trivalent at more alkaline pH and more divalent at more acid pH. Modelling of the citrate
species present as a function of pH (Barrow 2018) shows that important changes in Ca-citrate complexes only
occur in a pH range relevant to the acid soil in this study (4.5-5.5). The curves of $x_{32P}/[^{32}P]$ with increasing
carboxylate concentration followed the same trends at each acid addition. This implies that the anion effects, via
ligand exchange and chelation, dominates over pH. The magnitude of the carboxylate effect was slightly greater
for acidified soils, as seen by the convergence of curves.  The only exception is observed for the two calcareous
soils where there was an inversion of the trend with the highest oxalate concentration in the more acid suspension.
This may be attributed to precipitation of calcium oxalate (solubility product, $K_{SP} = 10^{-9}$ M), thus removing oxalate
from solution and reducing its interaction with soil P.
4. Conclusions
In conclusion, carboxylate anions addition led to marked increases in P lability assessed by isotopic
exchangeability, beyond a threshold of about 0.2 mmolc kg$^{-1}$ soil, with greater increases observed for oxalate than
citrate. Similar trends of carboxylate were observed when soils were simultaneously acidified. Acidification tended
to decrease P lability in all soils. In most cases the effect could be attributed to a salt effect (increase in ionic
strength). When calcareous soils were acidified by about 1 pH unit, the effect of oxalate was small, probably due
to precipitation of calcium oxalate. We found no evidence that P lability is greatest around neutral pH, although
this is often assumed to be the case.



**Figure and Table captions**
**Figure 1:** Distribution of $^{32}P$ on each of the soils, $x_{32P}/[^{32}P]$, as a function of added citrate or oxalate sodium salt
with or without acidification. Also shown is the distribution when chloride salt is added.
**Figure 2:** Comparison of the effect of acidification (ratio of $x/[P]$ measured at $pH_0$ and $pH_{-0.5}$) with a) initial soil
pH ($pH_0$); b) buffer capacity; and c) salt effect, without acidification (ratio of $x/[P]$ measured without NaCl and in
the presence of 0.1 N NaCl).
**Table 1:** Sampling locations and some properties of the four soils.



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




| Soil class | Sampling location | Code | Available P mg kg⁻¹ | CaCO₃ g kg⁻¹ | Corg mg kg⁻¹ | Texture | pH | pH buffer capacity mmol H⁺ kg⁻¹ soil pH⁻¹ |
|---|---|---|---|---|---|---|---|---|
| Chromic cambisol | Cazevieille, France | Cazevieille | 3.1 | <1 | 24.1 | Clay | 6.7 | 25.1 |
| Typic Hapludalf | Auzeville, Toulouse, France | Château Fourques | 2.8 | 92 | 7.7 | Silty clay | 7.8 | 74.9 |
| Luvisol | Girona, Spain | Montana Colera | 0.87 | 0 | 11.7 | Silt loam | 5.9 | 17.6 |
| Calcareous brown earth | Restinclière, France | Restinclière | 3.06 | 579 | 44.5 | Clay loam | 7.4 | 52.2 |

Table 1: **Sampling locations and some properties of the four soils.**



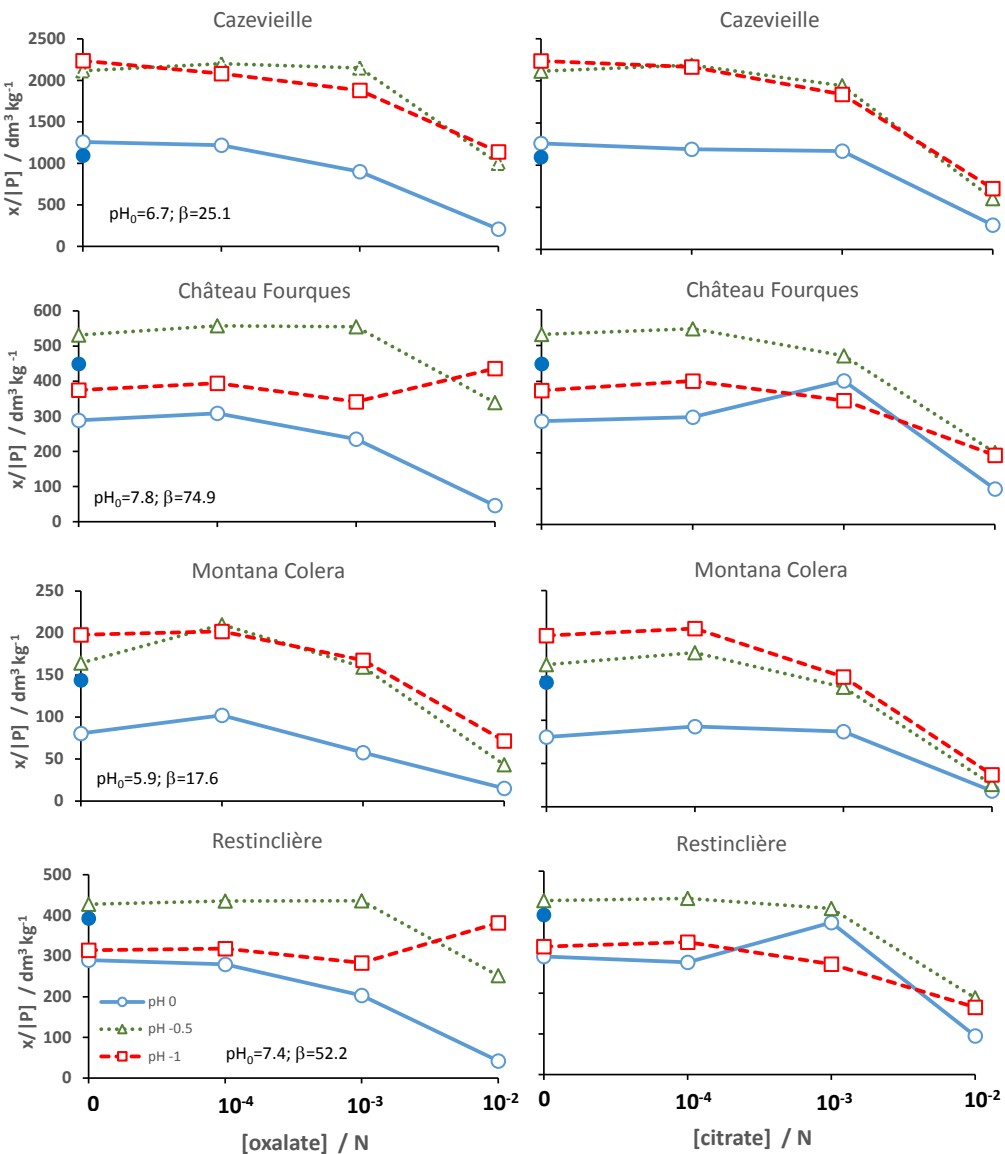

**Figure 1**: Distribution of $^{32}P$ on each of the soils, $x_{32P}/[^{32}P]$, as a function of added citrate or oxalate sodium salt with or without acidification. Also shown is the distribution when chloride salt is added (closed circle in blue). Values of initial pH and average buffer capacity ($\beta$) are given for each soil.



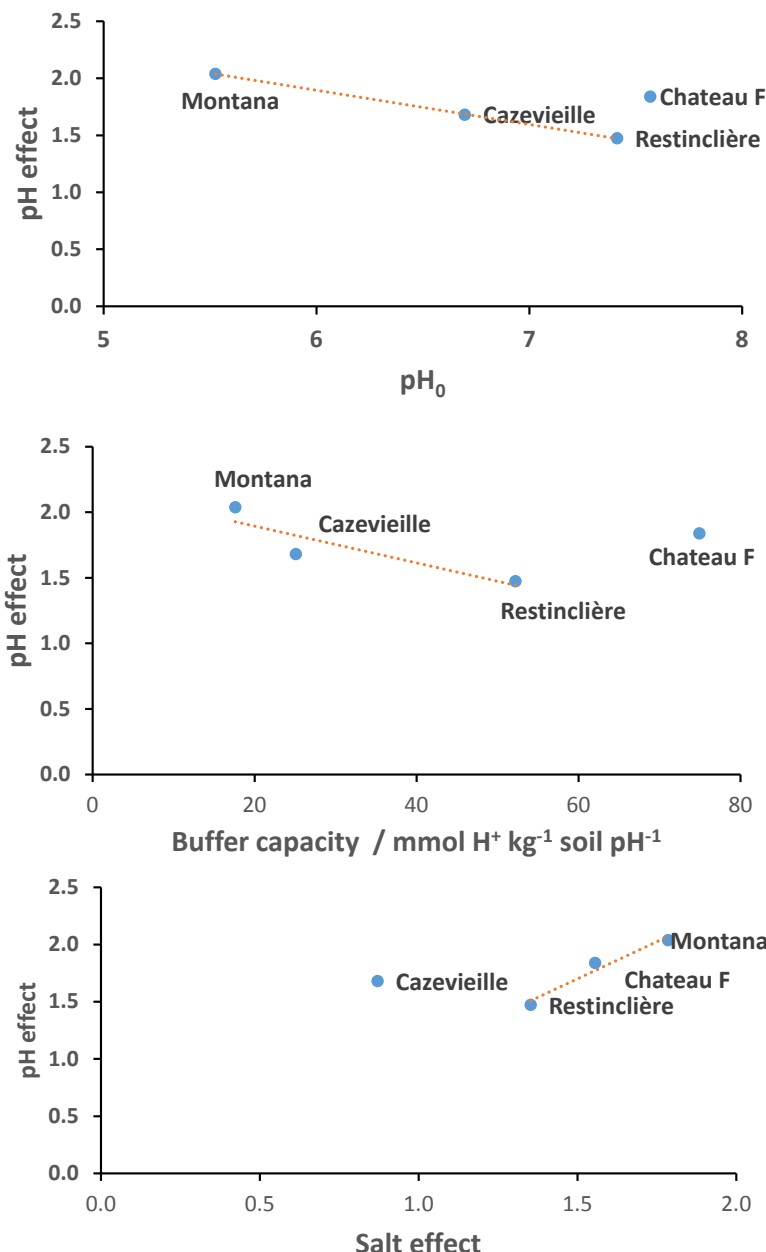

**Figure 2:** Comparison of the effect of acidification (ratio of x/[P] measured at pH0 and pH-0.5) with a) initial soil pH (pH0); b) buffer capacity; and c) salt effect, without acidification (ratio of x/[P] measured without NaCl and in the presence of 0.1 N NaCl).



**Declaration of interests**

The author declares that they have no known competing financial interests or personal relationships
that could have appeared to influence the work reported in this paper.



|  | Staunton | Pistocchi |
|---|---|---|
| 1. Conceptualization; | x | |
| 2. Data curation; | x | |
| 3. Formal analysis; | x | x |
| 4. Funding acquisition; | | |
| 5. Investigation; | x | |
| 6. Methodology; | x | x |
| 7. Project administration; | | |
| 8. Resources; | | |
| 9. Software; | | |
| 10. Supervision; | | |
| 11. Validation; | x | x |
| 12. Visualization; | | |
| 13. Roles/Writing - original draft; | x | x |
| 14. Writing - review & editing | x | x |