# Peer review of "Isotopic exchangeability reveals that soil phosphate is mobilised by"

_EGUsphere, 2024_

## Author Comment (AC1)

Revision of the manuscript: *Isotopic exchangeability reveals that soil phosphate is mobilised by carboxylate anions whereas acidification had the reverse effect.* Submitted to Egusphere by S. Staunton and C. Pistocchi.

We thank the reviewer for their careful reading and constructive comments. Our replies are indented below the text.

The work addresses the explanation of the mechanisms underpinning the release of phosphate from soil solid phases by the anions of organic acids commonly released in plant rhizosphere, in particular, oxalate and citrate.

The topic is not novel itself, being historically addressed by several publications of soil chemistry, (e.g., Barrow NJ (1984), J Soil Sci 35:283–297), some of which by the same Author (Staunton and Leprince (1996), Eur. J. Soil Sci., 47:231-239) as the Authors properly explain in their Introduction. However, the question is here afforded for the first time using 32P isotopic exchange, thus providing reliable information on P exchangeability; moreover, the experiment is designed to disentangle the contributions of the ionic strength of the solution, the changes in pH and the complexing/competing ability of the organic ions.

The experimental design is simple but smartly composed, well described and adequate for answering the research questions. The results are elaborated to evidence the differences among different soil samples and treatments induced by the variables of interest (They probably express the most suitable way to evidence the studied effects, however the addition of a table with the "raw data" od P exchange at each pH in the different soils would be nice to read). The different behaviors of the soils are not always easy to explain, however the given explanations are reliable.

> The data presented in Figure 1 have not been subjected to complex treatment. We understand that some readers may wish to reproduce or re-treat data, and thus we propose to supply raw data as supplementary information.

The writing style is clear and concise, in line the experimental setup and the data presentation in the tables and figures.

A few minor specific comments are reported below:

Page 2, line 24: "optimization of legacy soil P". May it be: "optimization of legacy soil P availability"?

> We have made this revision.

Page 5, line 128. Suggestion: could you add a short explanation about the mechanisms of "salt effect" vs "pH effect"?

> Thank you for this suggestion.

> The text has been clarified, as follows

Acidification by addition of HCl implies the addition of chloride and therefore any observed effect could be due to both salt concentration or acidification. Salt effects are associated with variable charge surfaces and the formation of cation bridges. The effects of pH (acidification) and salt addition ($10^{-2}$ N NaCl) are calculated by normalizing the ratio x/[P] with acidification (0.5) or salt (10 mM) with respect to that measured under standard conditions (no acidification or salt addition).

Table S1

Data shown in Figure 1. Ratio x/[P] for each of the soils with addition of various concentrations of sodium salt of oxalate, citrate or chloride with no acidification or acidification by 0.5 or 1 pH units.

| | | | | $x/[P]$ / dm$^3$ kg$^{-1}$ | | |
|---|---|---|---|---|---|---|
| $\Delta$pH | [salt] / N | Salt | Cazevieille | Château Fourques | Montana Colera | Restinclière |
| 0 | 0 | - | 1257 | 288 | 80 | 290 |
| -0.5 | 0 | - | 2115 | 531 | 164 | 427 |
| -1 | 0 | - | 2236 | 374 | 198 | 314 |
| 0 | $10^{-4}$ | Oxalate | 1219 | 308 | 102 | 280 |
| -0.5 | $10^{-4}$ | Oxalate | 2201 | 558 | 210 | 435 |
| -1 | $10^{-4}$ | Oxalate | 2081 | 394 | 202 | 319 |
| 0 | $10^{-3}$ | Oxalate | 900 | 235 | 58 | 203 |
| -0.5 | $10^{-3}$ | Oxalate | 2151 | 555 | 160 | 436 |
| -1 | $10^{-3}$ | Oxalate | 1880 | 342 | 168 | 283 |
| 0 | $10^{-2}$ | Oxalate | 209 | 46 | 15 | 42 |
| -0.5 | $10^{-2}$ | Oxalate | 1001 | 339 | 43 | 252 |
| -1 | $10^{-2}$ | Oxalate | 1140 | 436 | 72 | 382 |
| 0 | $10^{-4}$ | Citrate | 1189 | 300 | 93 | 276 |
| -0.5 | $10^{-4}$ | Citrate | 2188 | 547 | 178 | 433 |
| -1 | $10^{-4}$ | Citrate | 2166 | 401 | 206 | 325 |
| 0 | $10^{-3}$ | Citrate | 1168 | 401 | 87 | 374 |
| -0.5 | $10^{-3}$ | Citrate | 1942 | 471 | 138 | 408 |
| -1 | $10^{-3}$ | Citrate | 1840 | 346 | 150 | 272 |
| 0 | $10^{-2}$ | Citrate | 289 | 99 | 18 | 95 |
| -0.5 | $10^{-2}$ | Citrate | 602 | 201 | 26 | 188 |
| -1 | $10^{-2}$ | Citrate | 723 | 194 | 37 | 166 |
| 0 | $10^{-2}$ | Chloride | 1095 | 449 | 144 | 392 |
| -0.5 | $10^{-2}$ | Chloride | 2491 | 564 | 190 | 391 |
| -1 | $10^{-2}$ | Chloride | 2315 | 295 | 209 | 365 |

Table S2

Data shown in Figure 2. Initial soil pH (pH0), buffer capacity (mol H$^+$ kg$^{-1}$ soil $\Delta$pH$^{-1}$), pH effect (x/[P] with $\Delta$pH =0.5 with respect to that with no acidification) and salt effect (x/[P] with [chloride]=10$^{-2}$ N with respect to that with no salt addition).

| | Cazevieille | Château Fourques | Montana Colera | Restinclière |
|---|---|---|---|---|
| pH$_0$ | 6.70 | 7.57 | 5.53 | 7.41 |
| Buffer capacity | 15.94 | 5.34 | 22.72 | 7.66 |
| pH Effect | 1.84 | 1.84 | 2.04 | 1.47 |
| Salt Effect | 0.87 | 1.56 | 1.79 | 1.35 |

---

## Author Response (AR1)

Revision of the manuscript: *Isotopic exchangeability reveals that soil phosphate is mobilised by carboxylate anions whereas acidification had the reverse effect.* Submitted to SOIL by S. Staunton and C. Pistocchi.

We thank the reviewers for their careful reading and constructive comments. Our replies are indented below the text. Other minor changes have been made to the original text.

**Reviewer 1**

The work addresses the explanation of the mechanisms underpinning the release of phosphate from soil solid phases by the anions of organic acids commonly released in plant rhizosphere, in particular, oxalate and citrate.

The topic is not novel itself, being historically addressed by several publications of soil chemistry, (e.g., Barrow NJ (1984), J Soil Sci 35:283–297), some of which by the same Author (Staunton and Leprince (1996), Eur. J. Soil Sci., 47:231-239) as the Authors properly explain in their Introduction. However, the question is here afforded for the first time using 32P isotopic exchange, thus providing reliable information on P exchangeability; moreover, the experiment is designed to disentangle the contributions of the ionic strength of the solution, the changes in pH and the complexing/competing ability of the organic ions.

The experimental design is simple but smartly composed, well described and adequate for answering the research questions. The results are elaborated to evidence the differences among different soil samples and treatments induced by the variables of interest (They probably express the most suitable way to evidence the studied effects, however the addition of a table with the "raw data" od P exchange at each pH in the different soils would be nice to read). The different behaviors of the soils are not always easy to explain, however the given explanations are reliable.

> The data presented in Figure 1 have not been subjected to complex treatment. We understand that some readers may wish to reproduce or re-treat data, and thus we propose raw data as supplementary information.

The writing style is clear and concise, in line the experimental setup and the data presentation in the tables and figures.

A few minor specific comments are reported below:

Page 2, line 24: "optimization of legacy soil P". May it be: "optimization of legacy soil P availability"?

> We have made this revision.

Page 5, line 128. Suggestion: could you add a short explanation about the mechanisms of "salt effect" vs "pH effect"?

> Thank you for this suggestion.

> The text has been clarified, as follows

Acidification by addition of HCl implies the addition of chloride and therefore any observed effect could be due to both salt concentration or acidification. Salt effects are associated with variable charge surfaces and the formation of cation bridges. The effects of pH (acidification) and salt addition ($10^{-2}$ N NaCl) are calculated by normalizing the ratio x/[P] with acidification (0.5) or salt (10 mM) with respect to that measured under standard conditions (no acidification or salt addition).

**Reviewer 2**

This short communication describes some important research that represents a novel contribution to advancing understanding of key properties and processes that influence the fate and bioavailability of phosphorus in soil-plant systems. The rationale for the study, together with the methodology employed, were appropriate and clearly justified and described. The results as presented in Figure 1 clearly demonstrated that the presence of citrate and oxalate anions had a significant impact on the potential bioavailability of inorganic phosphorus adsorbed on soil oxide surfaces, and confirmed the role and importance of soil acidity. However, I simply could not comprehend the meaning or significance of the data presented in Figure 2, and strongly recommend that it be omitted from the manuscript.

The role of the presentation of our findings in Figure 2 was to unravel the complex interacting effects of pH, ionic strength and carboxylate anions. However we agree that is does not clarify our thinking. We have removed the figure. We have also redrawn Figure 1 and rearranged Table 1 to present data from soils in order of increasing pH, and buffer capacity.